# Updated Parameters for *Listeria monocytogenes* Dose–Response Model Considering Pathogen Virulence and Age and Sex of Consumer

**DOI:** 10.3390/foods13050751

**Published:** 2024-02-29

**Authors:** Régis Pouillot, Andreas Kiermeier, Laurent Guillier, Vasco Cadavez, Moez Sanaa

**Affiliations:** 1Independent Researcher, 18 rue Mohamed Al Ghazi, Rabat 10170, Morocco; 2Statistical Process Improvement Consulting and Training Pty Ltd., Gumeracha 5233, Australia; andreas.kiermeier@gmail.com; 3Risk Assessment Department, French Agency for Food, Environmental and Occupational Health & Safety (Anses), 14 rue Pierre et Marie Curie, 94701 Maisons-Alfort, France; 4Centro de Investigação de Montanha (CIMO), Campus de Santa Apolónia, Instituto Politécnico de Bragança, 5300-253 Bragança, Portugal; vcadavez@ipb.pt; 5Laboratório para a Sustentabilidade e Tecnologia em Regiões de Montanha, Campus de Santa Apolónia, Instituto Politécnico de Bragança, 5300-253 Bragança, Portugal; 6Nutrition and Food Safety Department, World Health Organization, 1211 Geneva, Switzerland

**Keywords:** *Listeria monocytogenes*, hazard characterization, dose–response, virulence, clonal complex

## Abstract

Better knowledge regarding the *Listeria monocytogenes* dose–response (DR) model is needed to refine the assessment of the risk of foodborne listeriosis. In 2018, the European Food Safety Agency (EFSA) derived a lognormal Poisson DR model for 14 different age–sex sub-groups, marginally to strain virulence. In the present study, new sets of parameters are developed by integrating the EFSA model for these sub-groups together with three classes of strain virulence characteristics (“less virulent”, “virulent”, and “more virulent”). Considering classes of virulence leads to estimated relative risks (RRs) of listeriosis following the ingestion of 1000 bacteria of “less virulent” vs. “more virulent” strains ranging from 21.6 to 24.1, depending on the sub-group. These relatively low RRs when compared with RRs linked to comorbidities described in the literature suggest that the influence of comorbidity on the occurrence of invasive listeriosis for a given exposure is much more important than the influence of the virulence of the strains. The updated model parameters allow better prediction of the risk of invasive listeriosis across a population of interest, provided the necessary data on population demographics and the proportional contribution of strain virulence classes in food products of interest are available. An R package is made available to facilitate the use of these dose–response models.

## 1. Introduction

*Listeria monocytogenes* poses a significant threat to public health due to its ability to cause severe infections, particularly in vulnerable populations. In 2022, 27 European Union (EU) member states reported 2738 confirmed invasive human cases of listeriosis, a notification rate of 0.62 cases per 100,000 population, with a fatality rate of 18.1% and a total of 286 deaths [1]. In the realm of food safety, understanding the dose–response (DR) relationship of *L. monocytogenes* is pivotal for accurate and specific risk assessment and risk management. 

In the context of microbiological food safety risk assessment, the DR model for any pathogen is the mathematical description of the relationship between the number of ingested pathogenic organisms and the probability of infection or illness. The DR relationship is poorly understood for human listeriosis. However, we know that this relationship is affected by the three aspects of the infectious disease triangle [2,3,4], namely, (i) the food matrix, (ii) host susceptibility, and (iii) pathogen characteristics/virulence. In practice, most DR models are developed from limited data and, to date, cannot fully incorporate all these effects. The DR model for *L. monocytogenes* is no exception in this respect [4].

The DR models developed by the FAO and WHO in 2004 [5] for *L. monocytogenes* were of the exponential form. In these DR models, the probability that the exposure to a single *L. monocytogenes* cell results in illness, the “*r*-value”, is considered independent of the presence of other cells, and is considered constant for all cells and all individuals of a given sub-population [2,6]. The response of individuals exposed to *L. monocytogenes* is highly variable [4]. However, in the original FAO and WHO [5] risk assessment for invasive listeriosis, the only variability considered in the DR model was that between the general and susceptible populations and this was accommodated by fitting different *r*-values for the two populations. This group of two models had been developed for *L. monocytogenes* in ready-to-eat (RTE) foods, based on the method described by Buchanan et al. [7]. The method involved fitting the DR model to surveillance data detailing *L. monocytogenes* contamination observed in the food supply and matching the expected versus actual total number of listeriosis cases annually per sub-population as closely as possible.

More recent DR models for *L. monocytogenes* were developed to better incorporate host susceptibility. Pouillot et al. [8] proposed a DR model (called the “lognormal Poisson model”) that considers the variability in the probability for a *L. monocytogenes* cell causing illness within a sub-population. Host susceptibility was considered using epidemiological surveillance data from France for 11 sub-populations that were defined according to underlying health conditions [8]. A similar approach was used by the EFSA and considered 14 age–sex sub-groups (age classes: 1–4, 5–14, 15–24, 25–44, 45–64, 65–74, and ≥75 years old and sex classes: male and female) as surrogates for underlying health conditions [9].

In contrast to the work on host susceptibility, Fritsch et al. [10] focused on the pathogen and differences in virulence. These researchers derived specific parameters of a lognormal Poisson model by incorporating parameters specific to three distinct categories of virulence, characterized using clonal complexes (CCs) and sequence types (STs). 

Finally, with respect to the food matrix, the FAO and WHO [11] concluded that not enough information was available to consider these aspects as part of the DR model.

While the Fritsch et al. [10] model evaluated the DR for three different classes of *Listeria* virulence marginally to the various sub-populations, the EFSA model [9] evaluated the DR model for 14 different sub-populations marginally to strain virulence. The objective of this study was to develop new sets of parameters for the lognormal Poisson model by integrating the EFSA model for age–sex sub-groups together with three classes of virulence characteristics as defined by Fritsch et al. [10], giving a total of 42 DR curves, one for each different age–sex and virulence class combination.

## 2. Materials and Methods

### 2.1. Dose–Response Model

The DR models we considered here link the ingested number of bacteria (as an actual number or as the mean of a Poisson distribution) with the marginal probability (over strains and over individuals of a given population) of developing an invasive case of listeriosis.

The DR model used by FAO and WHO [5] assumes that each ingested *L. monocytogenes* cell has a given and independent probability (*r*) of triggering invasive listeriosis. In this model, *r* is assumed to be constant within a given sub-population. Hence, the probability of developing invasive listeriosis following the ingestion of exactly *n* bacteria is
(1)$$Prob\left(\mathrm{ill}|r,n\right)=1-{\left(1-r\right)}^{n}$$
which is a “binomial DR model” as it follows a binomial process [12]. In contrast, if the number of bacteria is expressed as the mean of a Poisson distribution of parameter *d*, then the dose–response, integrated over the (serving-to-serving) variability for an average dose *d*, is written as
(2)$$Prob\left(\mathrm{ill}|r,d\right)=1-exp\left(-r\times d\right),$$
which is the “exponential DR model” [6]. FAO and WHO [5] inferred some sets of *r* parameters for two sub-populations and used, for the food examples described in their risk assessment, a median value for *r* of 1.06 × 10^−12^, which equals 10^−11.975^, for their defined “population with increased susceptibility” and 2.37 × 10^-14^, which equals 10^−13.625^, for their “healthy population” ([5], page 60). 

Pouillot et al. [8] revisited this DR model and considered a model where *r* is not constant but follows a lognormal (base 10) distribution. The marginal (over strains and individuals from a given sub-population) model can then be written as follows:(3)$$\mathrm{Prob}\left(\mathrm{ill}|n,\theta \right)=1-\int_{0}^{1}{\left(1-r\right)}^{n}f\left(r,\theta \;\right)dr,$$
with $f\left(r,\theta =\left\{\mu ,\sigma \right\}\right)$ being the density of the lognormal (*x* follows the lognormal (base 10) distribution with parameter *μ* and *σ* if log_10_(*x*) ~ Normal(*μ*, *σ*)) distribution with parameters *μ* (mean) and *σ* (standard deviation) and *n* being the exact number of bacteria ingested, giving the “lognormal binomial DR model”. Alternatively, if the dose *d* is expressed as the mean of a serving-to-serving Poisson distribution, then the result is the “lognormal Poisson DR model”, which is given by
(4)$$\mathrm{Prob}\left(\mathrm{ill}|n,\theta \right)=1-\int_{0}^{1}exp\left(-r\times d\right)f\left(r,\theta \;\right)dr$$

Note that the exponential DR model is a special case of the lognormal Poisson model with *σ* = 0. In Pouillot et al. [8], the parameters of the lognormal distributions for various sub-populations with underlying health conditions were estimated using Goulet et al.’s [13] data regarding relative risk of listeriosis in 11 sub-populations in France (pregnancy, non-hematological cancer, other cancers, etc.) and exposure data estimated from the United States. Note that the inference process considered no post-retail growth of *L. monocytogenes* and a maximal population density (MPD) of *L. monocytogenes* in products equals to 6.1 log_10_ cfu/g.

### 2.2. Inference Process

In the current work, Pouillot et al.’s [8] lognormal Poisson framework is applied to data originating from the EU, considering various classes of *L. monocytogenes* virulence. As stated above, the lognormal Poisson model has two parameters, i.e., the mean (*μ*) and the standard deviation (*σ*) of the normal distribution of the log_10_(*r*) parameter. Here, estimates of *μ* and *σ* are made for each combination of the exposed sub-population and the class of virulence that is being considered. The estimation process described by Pouillot et al. [8] involves estimating *σ* for each class of virulence, and then scaling *μ* to exposure and epidemiological data.

The following information is needed:A classification of *L. monocytogenes* strains as a function of their virulence;Estimates of the inter-individual variability in susceptibility within each sub-population and estimates of the inter-strain variability in virulence within each class of virulence;Estimates of the exposure of the considered sub-populations to the various classes of virulence;Estimates of the corresponding number of cases of invasive listeriosis for each combination of sub-populations/class of virulence.

The following sub-sections will develop how those estimates were obtained.

#### 2.2.1. Classification of Strain Virulence

We used the classes of virulence defined by Fritsch et al. ([10], their Figure 3). These categories were named “hypovirulence”, “medium virulence”, and “hypervirulence”. The concept of a virulence class was initially introduced for *L. monocytogenes* by Roche et al. in 2001 [14]. These authors initially categorized virulence into three classes, named “virulent”, “hypovirulent”, and “avirulent”. However, we contend that no strains of *L. monocytogenes* are completely avirulent. Therefore, we suggest renaming these classes as “less virulent”, “virulent”, and “more virulent”.” The term “hypovirulent” literally means “less virulent”, but we have chosen not to use it to prevent confusion with the terminology used by Roche et al. [14], who applied “hypovirulent” to what we consider the medium virulence category.

Fritsch et al. [10] used the clinical frequency data compiled by Maury et al. [15], who characterized 6633 *L. monocytogenes* strains isolated from clinical samples and food. A clinical frequency was calculated by dividing the number of clinical isolates of a particular CC by the total number of clinical and food isolates of that CC. Three groups of CCs or STs were then created using a dendrogram from those clinical frequencies. For CCs/STs not present in that list, Fritsch et al. [10] recommended classifying them in the “virulent” group (Table 1).

#### 2.2.2. Estimate of the Standard Deviation of log_10_(r) within a Sub-Population/Class of Virulence

Pouillot et al. [8] showed that the standard deviation of log_10_(*r*) for a given sub-population could be estimated from the inter-individual variability in the susceptibility, *σ_p_* for sub-population *p*, and the inter-strain variability of virulence, *σ_v_* for virulence class *v*. 

Similarly to Fritsch et al. [10] and EFSA [9], the estimate (*σ_p_* = 0.55 log_10_ cfu) of the inter-individual variability in the susceptibility for all sub-populations was used, as suggested by Pouillot et al. [8]. This estimate came from USFDA and FSIS assessments in which 90% of the individual variability within the population group with medium variability in susceptibility may be contained within a range of 1.8 log_10_ ([15], their Table IV-8). If *Q*_90,p_ is the log_10_ difference between the 5th and the 95th percentile of a lognormal distribution,
(5)$$\sigma_{p}=\frac{Q_{90,p}}{2\cdot \Phi^{-1}\left(0.95\right)}=0.55,$$
where $\Phi^{-1}$ denotes the inverse of the standard normal cumulative density function. For the inter-strain variability of virulence, the experimental data on mice reported in Fritsch et al. [10], collected from USFDA and FSIS ([15], their Table IV-3), were used. The median lethal doses (LD_50_) obtained by intraperitoneal infectious route in normal mice for 26 strains were collected from this table. The values were clustered in 3 different groups ([10], their Figure 2). From these, the estimates of *σ_v=lv_* = 1.12, *σ_v=v_* = 0.63, and *σ_v=mv_* = 0.52 log_10_ cfu were estimated for the “less virulent” (LD_50_ ranging from 6.80 to 9.70 log_10_ cfu), “virulent” (LD_50_ ranging from 4.49 to 6.23 log_10_ cfu), and “more virulent” (LD_50_ ranging from 2.57 to 3.67 log_10_ cfu) *L. monocytogenes* strains, respectively. Note that the larger standard deviation for the “less virulent” strains is related to the choice of distance cutoff for log_10_(r) used and incorporates two small but potentially heterogeneous virulence groups; these two groups were more variable than any observed in the other virulence classes.

These estimates of the inter-individual variability in the susceptibility and of the inter-strain variability of virulence can then be combined [8], assuming independence, e.g., for a sub-population consuming strains of the “more virulent” class: (6)$$\sigma_{v=mv,p}=\sqrt{\sigma_{p}^{2}+\sigma_{v=mv}^{2}}=\sqrt{0.55^{2}+0.52^{2}}=0.756.$$

#### 2.2.3. Proportion of Various Classes of Strain Virulence in EU RTE and Clinical Cases

Møller Nielsen et al. [16] selected a total of 1143 *L. monocytogenes* isolates, including 333 human clinical isolates and 810 isolates from the food chain. The food strains originated partially from the EU-wide baseline survey conducted in 2010 and 2011 [17]. The baseline survey collected representative samples of three types of RTE food: seafood (smoked and gravad fish), meats (packaged heat-treated meat products), and cheese and dairy (soft and semi-soft cheeses). Due to the lack of strains isolated from meat and cheeses, additional isolates from RTE meat products and cheeses were obtained from as many different EU member states as possible. 

Among the 576 food strains available in Annexes 1, 2, and 3 of that report, 290 strains from the baseline survey (Annex 1 in [16]) were selected for the seafood category (out of 294, 3 were excluded due to a lack of information on the sampling stage and 1 was excluded because the allelic type was not specified), 176 were retained for the meat product category, and 89 for the dairy products categories (i.e., all the strains from Annexes 1 and 2 in [16] under the categories “meat and meat products” and “milk and milk products”). Five strains isolated from vegetables (Annex 3 in [16]) were not considered due to the small number of strains. A total of 555 food strains from this study were eventually retained. Clinical isolates from assumed sporadic human cases collected during the baseline survey period, 2010–2011, were included in the study (262 strains).

The isolates were whole genome sequenced and their CC characterized. In Table 2, the proportion of each of these virulence classes estimated in seafood products, meat products, and cheese, as well as in sporadic cases of invasive listeriosis are presented. Note that the “others/unknown” category will be considered as “virulent”.

#### 2.2.4. Exposure of the EU Population to Various Classes of Virulence 

To estimate the exposure of the EU population to the various classes of virulence of *L. monocytogenes*, the exposure part of the generic quantitative microbial risk assessment (gQMRA) model of EFSA [9] was adapted. The EFSA model was based on that developed by Pérez-Rodriguez et al. [18].

In this model, the exposure to *L. monocytogenes* in the EU from various RTE food categories—including heat-treated meat, smoked and gravad fish, and soft and semi-soft cheeses—was estimated. Exposure was assessed for each of 14 sub-populations, i.e., male and female sub-population of individuals aged 1–4, 5–14, 15–24, 25–44, 45–64, 65–74, and ≥75 years of age. Note that no underlying health conditions, such as pregnancy or illnesses, were considered in this study. The process started at the retail stage and ended at consumption. For prevalence and concentration at retail, data from the EU-wide baseline survey was complemented with EU monitoring data and US data [19]. Food serving size and the number of servings per year were estimated from the EU food consumption database. Growth of *L. monocytogenes* was modeled from retail to consumption, using temperature–time profiles during transport and storage. The packaging conditions, i.e., reduced oxygen packaging (ROP, including both vacuum and modified atmosphere packaging) vs. normal packaging, were considered as a factor modifying the growth potential of *L. monocytogenes* in all RTE food sub-categories (except in soft and semi-soft cheese, for which no ROP was considered). MPD of 6.23 log_10_ cfu/g for cooked meat and sausages, 7.28 log_10_ cfu/g for cheeses, 7.29 log_10_ cfu/g for fish products, or 7.53 log_10_ cfu/g for pâté were used. The exposure was calculated at the end of the simulation model by multiplying the concentration by the serving size. The overall exposure to *L. monocytogenes* was then the sum of exposure from the consumption of 13 RTE foods (cold smoked fish, hot smoked fish, gravid fish, cooked meat, sausage, and pâté for normal packaging and ROP and soft and semi-soft cheese for normal packaging). The original EFSA model derived an empirical cumulative distribution function (ecdf) of the overall exposure to *L. monocytogenes* from consumption of RTE for each of the 14 sub-populations. See [9] for details and availability of the original code and data.

The R [20] code from EFSA [9] was adapted in this study to separate the estimated exposure to the various classes of strain virulence. For that purpose, the model was scaled to the frequencies of the various virulence classes as observed in Table 2. For example, the prevalence considered for cheese (13/3114) was multiplied by 12.4%, 47.2% + 7.9% = 55.1%, and 32.6% (see Table 2) to estimate the specific exposure to “less virulent”, “virulent”, and “more virulent” *L. monocytogenes* strains from cheese, respectively. Similar proportions were applied for all food categories. The model resulted in three empirical distribution functions (one per virulence class) for each of the 14 age–sex sub-populations. Note that this evaluation considered, as underlying assumptions, that the initial concentration of *L. monocytogenes* in contaminated food is independent of the virulence class, and that the general behavior, and growth in particular, of *L. monocytogenes* in all those products is also independent of the class of virulence.

In addition, the total number of eating occasions per year/sub-populations/considered RTE estimated by EFSA [9] from the EFSA consumption database was used.

#### 2.2.5. Prevalence of Invasive Listeriosis in the EU Population according to Classes of Virulence 

To estimate the number of invasive human listeriosis cases per sub-population for each class of virulence, the proportion of each class of virulence observed in sporadic cases in EU (Table 2) was applied to the number of invasive listeriosis cases in the EU during the 2008–2015 period for each of the 14 sub-population group, as estimated by EFSA ([9], their Table 1, page 28).

### 2.3. Scaling of the Model to Epidemiological Data 

These inferences led to: Estimates of the number of invasive human listeriosis cases in the EU from 2008 to 2015 within 14 sub-populations for 3 different classes of virulence (42 categories);An estimate of the prevalence of contaminated RTE and the ecdf of the exposure to *L. monocytogenes* from contaminated servings of RTE for each of these 42 categories;Estimates of the standard deviation of the lognormal (base 10) distribution of *r* for 3 classes of virulence, considered as similar in each sub-population for a given class of virulence.

With these data, the mean of the log_10_(*r*) population can be estimated to match the model outputs with the epidemiological surveillance data for the EU. The number of cases of invasive listeriosis linked to *L. monocytogenes* of the class of virulence *v* in population *C_p,v_* is actually [8]:(7)$$C_{p,v}={TEO}_{p}\times \pi_{p,v}\times \left(1-\int_{d=0}^{\infty }\int_{lr=-\infty }^{\infty }exp\left(-10^{lr}\times d\right)f_{p,v}\left(d\right)\varphi \left(lr;\mu_{p,v},\sigma_{v}\right)dlr\;dd\right),$$
where *p* is the index for the sub-population, *v* is the index for the virulence class, *TEO_p_* is the total eating occasions of RTE for population *p* during the 2008–2015 period (estimated as the annual total eating occasions for population *p* multiplied by 8 (years)), *π_p,v_* is the prevalence of contaminated RTE for population *p* and virulence class *v*, *lr* is the log_10_ of *r*, *f_p,v_* is the density function of the expected dose for population *p* and virulence class *v* for contaminated servings, and *φ*(*x*; *μ*, *σ*) is the probability density function of the normal distribution. In practice, the integration over *f_p,v_* is obtained by discretizing the ecdf by steps of 0.1 log_10_ cfu.

### 2.4. Implementation and Diffusion

Equation (7) has a single unknown, *μ_p,v_*, which can be estimated numerically using the integrate function from the R stats package. To transfer the DR models to users, a dedicated R package was developed. The code and data used to derive the model and the dedicated R package are available at www.github.com/rpouillot/FoodsDR (accessed on 1 February 2024).

## 3. Results

The combined standard deviation for the consuming sub-population and strain virulence were estimated to be 1.247 for the “less virulent”, 0.836 for the “virulent”, and 0.756 log_10_ cfu for the “more virulent” strains. The full set of parameters, namely, the mean and standard deviation of the lognormal distribution for *r* for each combination sub-population/class of virulence, is provided in Table 3.

Figure 1 illustrates the DR curves for doses ranging from 1 to 10^12^ bacteria (as a mean of the Poisson dose) for the EFSA model [9] that are to be compared with the DR obtained in this study. The FAO and WHO [5] DR for the healthy population (exponential dose–response with parameter 1.06 × 10^−12^) and for the increased susceptibility population (exponential dose–response with parameter 2.37 × 10^−14^) are provided for comparison. As expected, the “more virulent” strains lead to a DR model that is shifted to the left when compared to the “marginal” model (EFSA, [9]) or, even more, the “less virulent strains”.

The risk of invasive listeriosis for selected doses ranging from 100 bacteria to 1,000,000,000 bacteria using the FAO and WHO [5], the Pouillot et al. [8], the EFSA [9] and the model from this study are presented in Table 4. This table also reports, for an easier comparison, the relative risk (RR) from an exposure to 10^3^ bacteria when compared to the FAO and WHO [5] model, healthy population. Note that, from this table it is possible to obtain the RR of any model vs. another one by calculating the ratio of their respective RRs. As an example, the RR “this study, female > 75 years old—more virulent strains” vs. “this study, female > 75 years old—less virulent strains” is estimated to be 3669.7/152.3 = 24.1.

From Figure 1, and considering a dose of 10^3^ bacteria, which lies within the linear part of the DR curves, for calculating the RR, the following observations can be made: for the FAO and WHO [5] model, the RR between the general and susceptible populations is 44.7, with the latter resulting in the higher probability of illness of 1.06 × 10^−9^.

The RRs between the Pouillot et al. [8] and the FAO and WHO [5] DR display some differences. Indeed, for the less than 65-year-old individual with no underlying health conditions, the estimated risk is 344 times higher using the Pouillot et al. (2015) model than when using the FAO and WHO healthy population model and is still 7.68 times higher than the FAO and WHO increased susceptibility population model. When compared with the EFSA model [9], the Pouillot et al. [8] model estimates a risk for the less than 65-year-old individual with no underlying conditions, for a given dose of 10^3^ bacteria, that is 8.41 times higher than the risk estimated from the EFSA model for the least susceptible population (male, 15–24 years old). The Pouillot et al. [8] model is more “conservative” than the FAO and WHO [5] model or the EFSA one [9].

The RR between the least (males between 15 and 24 years old) and the most (males and females > 75 years old) susceptible EFSA populations was 31.5, with the latter resulting in the higher probability of illness of 3.04 × 10^−8^ from a dose of 10^3^ bacteria. The RRs between the least (males between 15 and 24 years old) and the most (males and females > 75 years old) susceptible EFSA populations and the general population of the FAO and WHO model were 40.8 and 1285, respectively. In contrast, RRs between the least (males between 15 and 24 years old) and the most (males and females > 75 years old) susceptible EFSA populations and the susceptible population of the FAO and WHO model were 0.91 and 28.7, respectively. These observations hold for the linear part of the DR model, i.e., doses up to about 10^8^. At greater doses (>10^11^), the RRs reduce, and may reverse, i.e., the FAO and WHO susceptible population model predicts higher probability of illness compared with the EFSA model. 

As explained above, the new DR model takes the three classes of strain virulence into account, in addition to the age–sex groups of the EFSA model, resulting in a total of 42 DR curves. Instead of displaying all of these, the least (males between 15 and 24 years old) and the most (females > 75 years old) susceptible EFSA populations were selected and displayed for each of the three virulence classes: “less virulent”, “virulent”, and “more virulent”. It should be noted that when integrating the virulence class information with the EFSA model, females over 75 years of age are marginally more susceptible than males of the same age. The resulting plot is shown in Figure 2.

As before, considering a dose of 10^3^ bacteria, which lies within the linear part of the DR curves, for calculating the RR, the following observations can be made from studying Figure 2 and Table 4. The RRs between the “less virulent” class and the “more virulent” class of *L. monocytogenes* strains depend on the population of interest, i.e., they are not constant. The RRs are 21.6 and 24.1 for the least susceptible population (males between 15 and 24 years old) and the most susceptible population (females > 75 years old), respectively. Comparing the general and susceptible population FAO/WHO model with the new DR model, and using the virulent class of strains, the following RRs can be calculated:(i)Susceptible FAO/WHO population versus the most susceptible population (females > 75 years old) yields an RR of 22.0;(ii)General FAO/WHO population versus least susceptible population (males between 15 and 24 years old) yields an RR of 50.1.(iii)The RR between the least susceptible group (males between 15 and 24 years old) exposed to the “less virulent” class of strains versus the most susceptible group (females > 75 years old) exposed to the most virulent class of strains is 655.1.(iv)Taking class of strain virulence and age–sex sub-populations into account results in a wider range of possible probability of illness at a specific dose (RR = 655) compared with the range obtained from the previous FAO and WHO model (RR = 45).

Overall, to compare the marginal DR over all sub-populations, we weighted the risk of invasive listeriosis for a 10^3^ dose according to the proportion of each sub-population as provided by the FAO and WHO ([5], 83% of healthy population and 17% of increased susceptibility population), Pouillot et al. ([8], their Table I, based on frequency of each sub-population in France), and the EFSA ([9], their Table 22, based on the number of eating occasions per sub-population). Using as a reference the FAO and WHO model [5], the overall RR for Pouillot et al.’s [8] model at a 10^3^ bacteria dose is estimated to be 576, the overall RR for the EFSA model [9] is 53, and the overall RR for this study model is 6 for the “less virulent” strains, 45 for the “virulent” strains, and 144 for the “more virulent” strains. 

## 4. Discussion

DR models are a critical component of the hazard characterization part of any food safety risk assessment, especially when they are quantitative [2]. For quantitative microbial risk assessments, they link the number of pathogenic organisms ingested by a consumer, as estimated at the end of the exposure assessment, with the probability of the health outcome of interest, usually infection or illness. Different mathematical forms for DR models are available, each with their own assumptions [6]. Nevertheless, so-called “single-hit” models, which assume that each pathogenic cell ingested has a probability of resulting in the health outcome of interests—the *r*-value—which is independent of all other pathogenic cells ingested, are the most common DR models. By incorporating additional assumptions and parameterizations for the *r*-value, increasingly complex models can be developed with the aim of better accounting for the food matrix, pathogen, and host characteristics of the infectious disease triangle [2].

In the case of *L. monocytogenes,* the health outcome of interest is generally invasive disease due to the associated sequalae and relatively high mortality rate [5]. The 2004 model used by the FAO and WHO [5] was of the exponential form, and broad differentiation of the host’s susceptibility—healthy versus susceptible—was incorporated through the use of two different *r*-values. However, FAO and WHO experts at the time recommended that “*More complete investigation of outbreaks and determination of the virulence characteristics of L. monocytogenes will make the dose-response relationships more accurate and precise*” [5].

Considerable effort has been expended over the last two decades in that regard and several different data sources can contribute to this [11]. Advances in refining the DR model for invasive listeriosis in humans have been as follows:Pouillot et al. [8] modeled the *r*-value as a log_10_-normal distribution. The host’s underlying health conditions, such as pregnancy, cancer, heart disease, etc., were incorporated using RRs for various sub-populations and a proportion of each sub-population estimated from the French population [13]. Exposure to *L. monocytogenes* was estimated from US data [19]. No growth from retail to consumption was considered. For the MPD, an important parameter in *L. monocytogenes* risk assessment [21], a value of 6.1 log_10_ cfu/g was used;Fritsch et al. [10] estimated *r*-values for 26 strains of *L. monocytogenes* and then grouped these strains into three classes of virulence according to the similarity of the *r*-value, namely, “hypovirulent” (called here “less virulent”), “medium virulent” (called here “virulent”), and “hypervirulent” (called here “more virulent”);The EFSA [9] estimated *r*-values for 14 age–sex combinations. Age and sex were used as surrogates for underlying health conditions because the demographic stratification into the 14 sub-populations was readily available across the EU, unlike actual information on underlying health conditions. The EFSA considered a bacterial growth from retail to consumption and MPDs ranging from 6.23 to 7.53 log_10_ cfu/g according to the considered RTE [9].In 2022, an FAO and WHO expert group [11] considered that the most appropriate DR model approaches would include variability in sub-population susceptibility *and* variability in strain virulence. The availability of an exposure model to *L. monocytogenes* in the EU [9] and of CC data from representative samples of food and clinical cases in the same population [16] made this derivation possible.The novelty of the current work is to conflate these three works to develop a DR model for invasive listeriosis that could consider differences in strain virulence class as well as better account for host susceptibility through the inclusion of surrogate information—age and sex. Consequently, this new model allows for more specific estimation of the risk of invasive listeriosis at the cost of two additional pieces of information, namely, population demographics and proportional contribution of strain virulence classes in food products of interest.

The assumption underlying the inference of the number of invasive human listeriosis cases per sub-population for each class of virulence was that the proportion of cases linked to “less virulent”, “virulent”, or “more virulent” strains is independent of age, sex, EU country of residence, underlying health conditions, etc. This assumption could be refined in the future if additional data regarding the cases of listeriosis based on age, sex, and country of residence become available.

It should be noted that the EFSA sub-groups, based on sex and age, were used as a surrogate for underlying health conditions [9]. This was linked to the fact that the EFSA data are more broadly representative geographically and that information about underlying health conditions was not available. Ideally, however, representative data on specific underlying health conditions, similar to those obtained from France [13], would be preferable; this would also allow the DR model to be more globally relevant. As an example, the difference between males and females for the classes of age “15–24 years old” and “25–44 years old” is most probably linked to pregnancy [9]. The DR model derived here may help to refine risk assessment but will not be able to really quantify the increased RRs of listeriosis for the most susceptible population, as was done by Pouillot et al. [8].

The major assumption regarding the derivation of this DR model is the assumption of a single lognormal distribution underlying the distribution of *r* parameters in each of the considered sub-populations/classes of virulence. As described above, the use of highly heterogeneous sub-populations (pregnant and non-pregnant women for the female 15–24 and 25–44-year-old groups, individuals with severe underlying health conditions, and healthy individuals for groups of aged individuals) may impair this assumption. The other assumptions used in this derivation are all the various assumptions used in the EFSA generic quantitative microbial risk assessment model [9], notably the fact that all listeriosis cases were assumed to be linked to the considered limited number of RTE foods. This study makes the additional assumption of a similar behavior of the strains in this generic quantitative microbial risk assessment irrespective of their class of virulence.

Our estimates for μ and σ are simple point estimates. The uncertainty surrounding this DR model combines all the uncertainty linked to the exposure model, the estimated number of eating occasions, the reported number of clinical cases per sub-population, and the estimated proportions of classes of virulence in food and clinical cases. Regarding the exposure model and the number of eating occasions, the original publications ([18], their Table 25 and [9], their table J.1) characterized, documented, and explained all types of uncertainty, following the EFSA’s recommendations [22]. All the assumptions and limitations identified in these reports apply here. However, no quantification of the uncertainty was performed in these studies. The EFSA [9] performed an importance analysis for some parameters that they considered as the most influential, i.e., regarding duration and temperature of storage in the consumer’s refrigerator and MPD. Those parameters are considered to have the largest effect in risk assessments linked to *L. monocytogenes* in RTE foods [15,21,23]. The EFSA [9] found that the risk assessment model was very sensitive to a shift in the *L. monocytogenes* MPD, as expected [21]. We re-ran the model using a shift of +0.5 log_10_ of the MPD of *L. monocytogenes* for all RTE foods and observed that the estimated risk of invasive listeriosis following the ingestion of 1000 bacteria would be multiplied by a factor of 2.5 to 2.6, according to the sub-group and class of virulence. 

The model is additionally scaled to the number of clinical cases estimated for the 42 sub-groups. Some inferences were made based on a very small, estimated number of cases. For example, inferences for females aged 1–4 years old exposed to the “less virulent” strains were made from only four reported listeriosis cases. More uncertainty could be found surrounding these lower risk groups.

Although some CCs are present globally and are very similar at the core-genome level, their accessory genome can vary from region to region [24]. Within these variations in the accessory, some genetic elements may affect virulence. Regional differences in the accessory genome could then lead to variations in virulence within a given CC, which may not be captured in this study based on data from a single geographic region. 

Moreover, limited data were available to classify some CCs. For instance, the proportion of CC87 strains in the EU is quite small. The classification of CC87 in the “more virulent” category was based only on 10 strains in foods and 4 in sporadic cases. Using data from China could provide more information as CC87 strains are predominant in Chinese food isolates and in sporadic clinical infections [25]. This points to the challenge of classifying virulence based on limited data and suggests that incorporating more comprehensive datasets, especially from regions where certain strains are more prevalent, could lead to more accurate classifications.

Collectively this model suggests higher *r* values than those derived previously by the FAO and WHO [5], but lower than the more conservative model from Pouillot et al. [8]. The effect of higher *r* values is that they result in a left shift of the DR curve, i.e., to lower doses. Considering classes of virulence leads to RRs between “less virulent” and “more virulent” strains ranging from 21.6 to 24.1. These relatively low RRs should be compared to the extremely high RRs obtained by Goulet et al. for various comorbidities [13] (e.g., 373.6 for hematological cancer when compared to the less than 65-year-old individual with no known underlying conditions). This observation means that the influence of comorbidity seems much more important than the influence of the strains, as estimated in this study. 

## 5. Conclusions

Parameters for lognormal Poisson DR models for *L. monocytogenes* were estimated for three classes of strain virulence and 14 age–sex sub-populations, as a surrogate for underlying health conditions. The updated model parameters allow better estimation of the risk of invasive listeriosis across a population of interest, provided the necessary data on population demographics and the proportional contribution of strain virulence classes in food products of interest are available. In turn, these improved estimates will facilitate better decision making in food safety risk assessments, though future focus on comorbidity data seems much more important than data on strain virulence variability.

## Figures and Tables

**Figure 1 foods-13-00751-f001:**
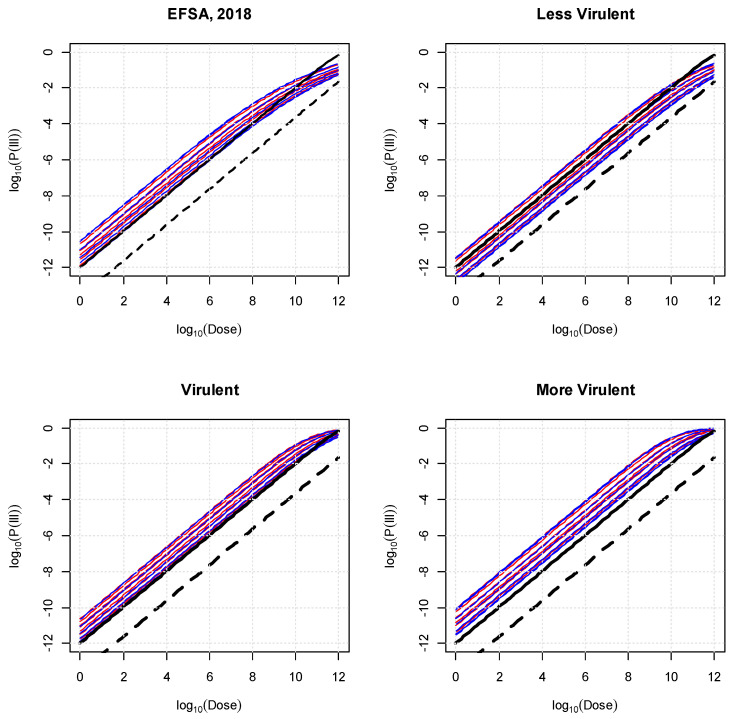
Dose–response curves for 14 age–sex groups (females = red lines and males = blue lines) from EFSA [9] (top, left) and this study (“less virulent”, “virulent”, and “more virulent” strains). FAO and WHO [5] models for the general (black dashed line) and susceptible (black solid line) populations.

**Figure 2 foods-13-00751-f002:**
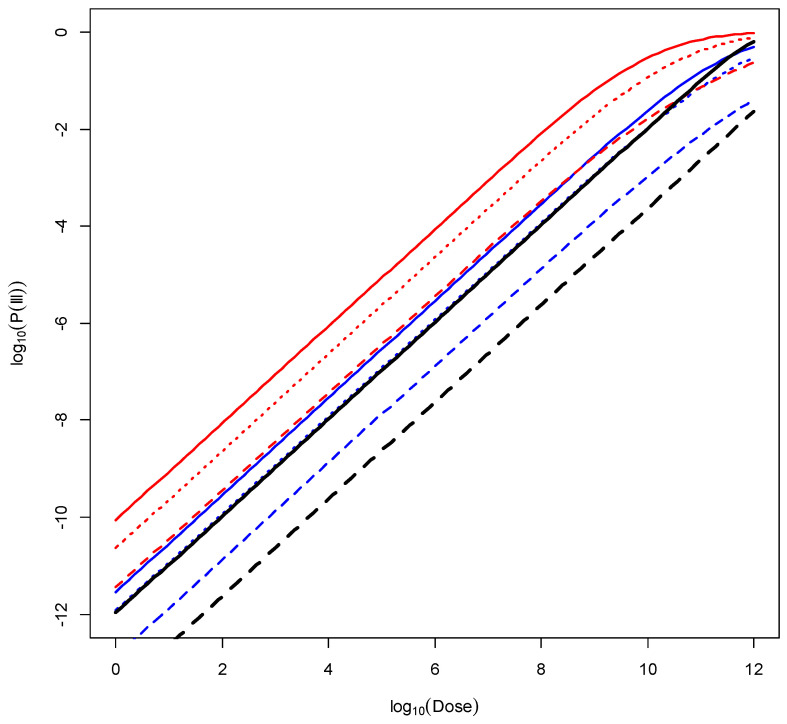
DR model curves for the least susceptible population (blue, males between 15 and 24 years old) and the most susceptible population (red, females > 75 years old); “less virulent” class of strains (dashed red/blue lines), “virulent” (dotted red/blue lines), and “more virulent” (solid red/blue lines); and FAO and WHO [5] susceptible population (black solid line) and general population (black dashed line).

**Table 1 foods-13-00751-t001:** Classification of strains in three classes of virulence (from [10]).

Less Virulent Strains	Virulent Strains	More Virulent Strains
CC121, CC204, CC31, CC9, CC193, CC19, ST214	CC14, CC155, CC177, CC18, CC20, CC21, CC26, CC3, CC37, CC379, C388, CC398, CC5, CC59, CC8, CC403 and all others	CC1, CC101, CC2, CC220, CC224, CC4, CC451, CC54, CC6, CC7, CC87

**Table 2 foods-13-00751-t002:** Proportion of the different classes of virulence in seafood, meats, and cheese and dairy, and in sporadic cases in the EU from [16].

	Less Virulent	Virulent	More Virulent	Others/Unknown	*n*
RTE Seafood	149 (51.4%)	102 (35.2%)	36 (12.4%)	3 (1.0%)	290
RTE Meats	104 (59.1%)	35 (19.9%)	35 (19.9%)	2 (1.1%)	176
RTE cheese and dairy	11 (12.4%)	42 (47.2%)	29 (32.6%)	7 (7.9%)	89
Sporadic	22 (8.4%)	76 (29.0%)	156 (59.5%)	8 (3.1%)	262

**Table 3 foods-13-00751-t003:** Mean (*μ*) and standard deviation (*σ*) of the lognormal (base 10) distributions of *r* for the 14 sub-populations and the 3 classes of virulence. The arithmetic mean of *r* is given in parenthesis.

Population	Less Virulent (*σ* = 1.247)	Virulent (*σ* = 0.836)	More Virulent (*σ* = 0.756)
Female 1–4 yo	−14.166 (4.22 × 10^−13^)	−12.296 (3.22 × 10^−12^)	−11.671 (9.71 × 10^−12^)
Male 1–4 yo	−14.124 (4.65 × 10^−13^)	−12.256 (3.54 × 10^−12^)	−11.625 (1.08 × 10^−11^)
Female 5–14 yo	−14.516 (1.89 × 10^−13^)	−12.582 (1.67 × 10^−12^)	−12.046 (4.09 × 10^−12^)
Male 5–14 yo	−14.633 (1.44 × 10^−13^)	−12.690 (1.30 × 10^−12^)	−12.165 (3.11 × 10^−12^)
Female 15–24 yo	−14.002 (6.45 × 10^−13^)	−12.123 (4.80 × 10^−12^)	−11.521 (1.37 × 10^−11^)
Male 15–24 yo	−14.668 (1.33 × 10^−13^)	−12.730 (1.19 × 10^−12^)	−12.200 (2.87 × 10^−12^)
Female 25–44 yo	−13.708 (1.21 × 10^−12^)	−11.815 (9.76 × 10^−12^)	−11.239 (2.63 × 10^−11^)
Male 25–44 yo	−14.444 (2.22 × 10^−13^)	−12.522 (1.92 × 10^−12^)	−11.973 (4.84 × 10^−12^)
Female 45–64 yo	−13.755 (1.09 × 10^−12^)	−11.890 (8.21 × 10^−12^)	−11.272 (2.43 × 10^−11^)
Male 45–64 yo	−13.753 (1.09 × 10^−12^)	−11.869 (8.61 × 10^−12^)	−11.274 (2.42 × 10^−11^)
Female 65–74 yo	−13.418 (2.36 × 10^−12^)	−11.594 (1.62 × 10^−11^)	−10.916 (5.52 × 10^−11^)
Male 65–74 yo	−13.283 (3.23 × 10^−12^)	−11.447 (2.28 × 10^−11^)	−10.785 (7.45 × 10^−11^)
Female >75 yo	−13.234 (3.61 × 10^−12^)	−11.437 (2.33 × 10^−11^)	−10.718 (8.70 × 10^−11^)
Male >75 yo	−13.255 (3.44 × 10^−12^)	−11.468 (2.17 × 10^−11^)	−10.734 (8.38 × 10^−11^)

**Table 4 foods-13-00751-t004:** Risk of invasive listeriosis for selected doses according to the dose–response model and the population. Relative risk from a dose of 10^3^ bacteria (reference: FAO and WHO model, “healthy population” [5]).

Model	Population	Virulence	Dose (Bacteria)				Relative Risk
			10^2^	10^3^	10^6^	10^9^	
FAO and WHO [5]	Healthy Population	All strains	2.37 × 10^−12^	2.37 × 10^−11^	2.37 × 10^−08^	2.37 × 10^−05^	Reference
	Increased susceptibility		1.06 × 10^−10^	1.06 × 10^−09^	1.06 × 10^−06^	1.06 × 10^−03^	44.7
Pouillot et al. [8]	Less than 65 yo, no underlying condition	All strains	8.15 × 10^−10^	8.14 × 10^−09^	7.33 × 10^−06^	2.74 × 10^−03^	343.5
	More than 65 yo, no underlying condition		1.55 × 10^−08^	1.53 × 10^−07^	1.10 × 10^−04^	1.94 × 10^−02^	6473.8
	Pregnancy		2.06 × 10^−07^	1.98 × 10^−06^	9.73 × 10^−04^	7.70 × 10^−02^	83,675.8
	Non-hematological cancer		8.09 × 10^−08^	7.89 × 10^−07^	4.54 × 10^−04^	4.85 × 10^−02^	33,270.7
	Hematological Cancer		9.66 × 10^−07^	8.93 × 10^−06^	3.18 × 10^−03^	1.50 × 10-01	376,694.8
	Renal or Liver Failure		2.84 × 10^−07^	2.71 × 10^−06^	1.25 × 10^−03^	8.93 × 10^−02^	114,387.0
	Solid Organ Transplant		3.18 × 10^−07^	3.03 × 10^−06^	1.37 × 10^−03^	9.40 × 10^−02^	127,857.9
	Inflammatory diseases		8.66 × 10^−08^	8.44 × 10^−07^	4.81 × 10^−04^	5.03 × 10^−02^	35,604.0
	HIV/AIDS		6.73 × 10^−08^	6.58 × 10^−07^	3.90 × 10^−04^	4.41 × 10^−02^	27,763.9
	Diabetes		7.78 × 10^−09^	7.72 × 10^−08^	5.96 × 10^−05^	1.27 × 10^−02^	3259.4
	Heart Diseases		5.26 × 10^−09^	5.23 × 10^−08^	4.18 × 10^−05^	9.90 × 10^−03^	2207.8
EFSA [9]	Female 1–4 yo	All strains	2.80 × 10^−10^	2.80 × 10^−09^	2.63 × 10^−06^	1.22 × 10^−03^	118.1
	Male 1–4 yo		3.58 × 10^−10^	3.58 × 10^−09^	3.33 × 10^−06^	1.48 × 10^−03^	151.1
	Female 5–14 yo		1.27 × 10^−10^	1.27 × 10^−09^	1.22 × 10^−06^	6.54 × 10^−04^	53.8
	Male 5–14 yo		1.04 × 10^−10^	1.04 × 10^−09^	9.99 × 10^−07^	5.54 × 10^−04^	43.8
	Female 15–24 yo		4.97 × 10^−10^	4.97 × 10^−09^	4.57 × 10^−06^	1.89 × 10^−03^	209.5
	Male 15–24 yo		9.67 × 10^−11^	9.67 × 10^−10^	9.32 × 10^−07^	5.23 × 10^−04^	40.8
	Female 25–44 yo		9.92 × 10^−10^	9.90 × 10^−09^	8.83 × 10^−06^	3.15 × 10^−03^	417.7
	Male 25–44 yo		1.81 × 10^−10^	1.81 × 10^−09^	1.72 × 10^−06^	8.66 × 10^−04^	76.3
	Female 45–64 yo		8.72 × 10^−10^	8.70 × 10^−09^	7.81 × 10^−06^	2.87 × 10^−03^	367.2
	Male 45–64 yo		9.47 × 10^−10^	9.45 × 10^−09^	8.45 × 10^−06^	3.05 × 10^−03^	398.9
	Female 65–74 yo		2.09 × 10^−09^	2.08 × 10^−08^	1.78 × 10^−05^	5.33 × 10^−03^	877.5
	Male 65–74 yo		2.89 × 10^−09^	2.88 × 10^−08^	2.41 × 10^−05^	6.66 × 10^−03^	1215.8
	Female > 75 yo		3.06 × 10^−09^	3.04 × 10^−08^	2.54 × 10^−05^	6.92 × 10^−03^	1284.7
	Male > 75 yo		3.06 × 10^−09^	3.04 × 10^−08^	2.54 × 10^−05^	6.92 × 10^−03^	1284.7
This Study	Female 1–4 yo	Less virulent	4.22 × 10^−11^	4.22 × 10^−10^	4.22 × 10^−07^	3.77 × 10^−04^	17.8
	Male 1–4 yo		4.65 × 10^−11^	4.65 × 10^−10^	4.65 × 10^−07^	4.12 × 10^−04^	19.6
	Female 5–14 yo		1.89 × 10^−11^	1.89 × 10^−10^	1.88 × 10^−07^	1.75 × 10^−04^	8.0
	Male 5–14 yo		1.44 × 10^−11^	1.44 × 10^−10^	1.44 × 10^−07^	1.35 × 10^−04^	6.1
	Female 15–24 yo		6.15 × 10^−11^	6.15 × 10^−10^	6.14 × 10^−07^	5.35 × 10^−04^	26.0
	Male 15–24 yo		1.33 × 10^−11^	1.33 × 10^−10^	1.33 × 10^−07^	1.25 × 10^−04^	5.6
	Female 25–44 yo		1.21 × 10^−10^	1.21 × 10^−09^	1.21 × 10^−06^	9.96 × 10^−04^	51.2
	Male 25–44 yo		2.22 × 10^−11^	2.22 × 10^−10^	2.22 × 10^−07^	2.05 × 10^−04^	9.4
	Female 45–64 yo		1.09 × 10^−10^	1.09 × 10^−09^	1.09 × 10^−06^	9.02 × 10^−04^	45.9
	Male 45–64 yo		1.09 × 10^−10^	1.09 × 10^−09^	1.09 × 10^−06^	9.05 × 10^−04^	46.0
	Female 65–74 yo		2.36 × 10^−10^	2.36 × 10^−09^	2.36 × 10^−06^	1.80 × 10^−03^	99.7
	Male 65–74 yo		3.23 × 10^−10^	3.23 × 10^−09^	3.21 × 10^−06^	2.37 × 10^−03^	136.1
	Female > 75 yo		3.61 × 10^−10^	3.61 × 10^−09^	3.59 × 10^−06^	2.61 × 10^−03^	152.3
	Male > 75 yo		3.44 × 10^−10^	3.44 × 10^−09^	3.42 × 10^−06^	2.50 × 10^−03^	144.9
	Female 1–4 yo	Virulent	3.22 × 10^−10^	3.22 × 10^−09^	3.22 × 10^−06^	3.08 × 10^−03^	136.0
	Male 1–4 yo		3.54 × 10^−10^	3.54 × 10^−09^	3.54 × 10^−06^	3.37 × 10^−03^	149.2
	Female 5–14 yo		1.67 × 10^−10^	1.67 × 10^−09^	1.67 × 10^−06^	1.63 × 10^−03^	70.4
	Male 5–14 yo		1.30 × 10^−10^	1.30 × 10^−09^	1.30 × 10^−06^	1.27 × 10^−03^	54.9
	Female 15–24 yo		4.80 × 10^−10^	4.80 × 10^−09^	4.80 × 10^−06^	4.52 × 10^−03^	202.6
	Male 15–24 yo		1.19 × 10^−10^	1.19 × 10^−09^	1.19 × 10^−06^	1.17 × 10^−03^	50.1
	Female 25–44 yo		9.76 × 10^−10^	9.76 × 10^−09^	9.76 × 10^−06^	8.80 × 10^−03^	412.0
	Male 25–44 yo		1.92 × 10^−10^	1.92 × 10^−09^	1.92 × 10^−06^	1.86 × 10^−03^	80.9
	Female 45–64 yo		8.21 × 10^−10^	8.21 × 10^−09^	8.20 × 10^−06^	7.48 × 10^−03^	346.2
	Male 45–64 yo		8.61 × 10^−10^	8.61 × 10^−09^	8.61 × 10^−06^	7.83 × 10^−03^	363.3
	Female 65–74 yo		1.62 × 10^−09^	1.62 × 10^−08^	1.62 × 10^−05^	1.40 × 10^−02^	685.3
	Male 65–74 yo		2.28 × 10^−09^	2.28 × 10^−08^	2.28 × 10^−05^	1.89 × 10^−02^	960.5
	Female > 75 yo		2.33 × 10^−09^	2.33 × 10^−08^	2.33 × 10^−05^	1.93 × 10^−02^	982.4
	Male > 75 yo		2.17 × 10^−09^	2.17 × 10^−08^	2.17 × 10^−05^	1.81 × 10^−02^	916.1
	Female 1–4 yo	More Virulent	9.71 × 10^−10^	9.71 × 10^−09^	9.71 × 10^−06^	9.06 × 10^−03^	409.6
	Male 1–4 yo		1.08 × 10^−09^	1.08 × 10^−08^	1.08 × 10^−05^	9.99 × 10^−03^	454.5
	Female 5–14 yo		4.09 × 10^−10^	4.09 × 10^−09^	4.09 × 10^−06^	3.95 × 10^−03^	172.4
	Male 5–14 yo		3.11 × 10^−10^	3.11 × 10^−09^	3.11 × 10^−06^	3.03 × 10^−03^	131.2
	Female 15–24 yo		1.37 × 10^−09^	1.37 × 10^−08^	1.37 × 10^−05^	1.25 × 10^−02^	578.3
	Male 15–24 yo		2.87 × 10^−10^	2.87 × 10^−09^	2.87 × 10^−06^	2.80 × 10^−03^	121.2
	Female 25–44 yo		2.63 × 10^−09^	2.63 × 10^−08^	2.62 × 10^−05^	2.27 × 10^−02^	1107.6
	Male 25–44 yo		4.84 × 10^−10^	4.84 × 10^−09^	4.84 × 10^−06^	4.65 × 10^−03^	204.0
	Female 45–64 yo		2.43 × 10^−09^	2.43 × 10^−08^	2.43 × 10^−05^	2.12 × 10^−02^	1026.1
	Male 45–64 yo		2.42 × 10^−09^	2.42 × 10^−08^	2.42 × 10^−05^	2.10 × 10^−02^	1020.0
	Female 65–74 yo		5.52 × 10^−09^	5.52 × 10^−08^	5.52 × 10^−05^	4.33 × 10^−02^	2330.5
	Male 65–74 yo		7.45 × 10^−09^	7.45 × 10^−08^	7.45 × 10^−05^	5.55 × 10^−02^	3145.5
	Female > 75 yo		8.70 × 10^−09^	8.70 × 10^−08^	8.69 × 10^−05^	6.28 × 10^−02^	3669.7
	Male > 75 yo		8.38 × 10^−09^	8.38 × 10^−08^	8.37 × 10^−05^	6.10 × 10^−02^	3536.8

## Data Availability

Data, code and an R package are available at www.github.com/rpouillot/FoodsDR (accessed on 1 February 2024).

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
