# Peer review of "Updated Parameters for Listeria monocytogenes Dose–Response Model Considering Pathogen Virulence and Age and Sex of Consumer"

_foods, 2024, doi:10.3390/foods13050751_

Round 1

Reviewer 1 Report

Comments and Suggestions for Authors

The idea of “three classes of strain virulence characteristics (“less virulent”, “virulent”, “more virulent”)” is interesting and necessary for the updated DRM in QMRA system. The updated model parameters allow better prediction of the risk of invasive listeriosis across a 29 population of interest, provided the necessary data on population demographics and the proportional contribution of strain virulence classes in food products of interest are available.

L47 “DR models are developed from limited data and cannot fully incorporate all these 47 effects.” check this and it’s a strong statement. What about MRA 4/5/38 progress?

L85, what’s the reference for this definition or explanation of DRM?

The whole analysis seems to be based on European conditions, so what’s about the differences with global or developing countries? E.g. Different countries or regions have different CC/ST dominate or typical groups.

Author Response

Reviewer 1

The idea of “three classes of strain virulence characteristics (“less virulent”, “virulent”, “more virulent”)” is interesting and necessary for the updated DRM in QMRA system. The updated model parameters allow better prediction of the risk of invasive listeriosis across a 29 population of interest, provided the necessary data on population demographics and the proportional contribution of strain virulence classes in food products of interest are available.

Thank you

L47 “DR models are developed from limited data and cannot fully incorporate all these 47 effects.” check this and it’s a strong statement. What about MRA 4/5/38 progress?

MRA 4/5/38 provides generalities on the subject but does not actually "incorporate all effects" in a mathematical model. We modified the manuscript "DR models are developed from limited data and, to date, cannot fully incorporate all these effects"

L85, what’s the reference for this definition or explanation of DRM?

We changed the manuscript in "The DR models we considered here link the ingested number of bacteria (as an actual number or as the mean of a Poisson distribution) with the marginal probability (over strains and over individuals of a given population) of developing an invasive case of listeriosis."

The whole analysis seems to be based on European conditions, so what’s about the differences with global or developing countries? E.g. Different countries or regions have different CC/ST dominate or typical groups.

We discuss this in the discussion section "Although some CCs are present globally and are very similar at the core-genome level, their accessory genome can vary from region to region [24]. Within these variations in the accessory, some genetic elements may affect virulence. Regional differences in the accessory genome could then lead to variations in virulence within a given CC, which may not be captured in this study based on data from a single geographic region.

Moreover, limited data were available to classify some CCs. For instance, the proportion of CC87 strains in the EU is quite small. The classification of CC87 in the “more virulent” category was based only on 10 strains in foods and 4 in sporadic cases. Using data from China could provide more information, as CC87 strains are predominant in Chinese food isolates and in sporadic clinical infections [25]. This points to the challenge of classifying virulence based on limited data and suggests that incorporating more comprehensive datasets, especially from regions where certain strains are more prevalent, could lead to more accurate classifications."

Reviewer 2 Report

Comments and Suggestions for Authors Updated parameters for the dose-response model for Listeria monocytogenes considering pathogen virulence and age and sex of consumer.   The above-said MS is well written by the authors and justified with the appropriate model.   Few things where I need the author's attention. 1. For this study no ethical permission is required. 2. Criteria of sub-population selection (any statistical tool used). 3. In the introduction section authors must mention the prevalence of listeriosis in the past with some death cases, especially in the EU.

Author Response

Updated parameters for the dose-response model for Listeria monocytogenes considering pathogen virulence and age and sex of consumer.   The above-said MS is well written by the authors and justified with the appropriate model.  

Thank you

Few things where I need the author's attention. 1. For this study no ethical permission is required.

No. Actually, no new data was collected for this study.

2. Criteria of sub-population selection (any statistical tool used).

As indicated in the manuscript, we followed the EFSA classification, without further discussion.

3. In the introduction section authors must mention the prevalence of listeriosis in the past with some death cases, especially in the EU.

We added "In 2022, the European Union (EU) Member States reported 2,738 confirmed invasive human cases of listeriosis, that is 0.62 cases per 100,000 population, with a fatality rate of 18.1% for a total of 286 deaths [1]. "

Using the reference

  1. European Food Safety Authority (EFSA); European Centre for Disease Prevention and Control (ECDC) The European Union One Health 2022 Zoonoses Report. EFSA Journal 2023, 21, e8442, doi:10.2903/j.efsa.2023.8442.